Effects of reduced dissolved oxygen concentrations on physiology and fluorescence of hermatypic corals and benthic algae

Haas Andreas F. 1 2 ahaas@ucsd.edu
Smith Jennifer E. 2
Thompson Melissa 2
Deheyn Dimitri D. 2 ddeheyn@ucsd.edu
1 Department of Biology, San Diego State University , United States
2 Scripps Institution of Oceanography, University of California , San Diego , United States
Qian Pei-Yuan
Electronic publication date: 2014 Jan 2
Publication date: 2014
Volume: 2
Electronic Location ID: e235
Received 2013 Oct 1; Accepted 2013 Dec 11
Copyright: © 2014 Haas et al.
Copyright year: 2014
Copyright holder: Haas et al.
License: This is an open access article distributed under the terms of the Creative Commons Attribution License, which permits unrestricted use, distribution, and reproduction in any medium, provided the original author and source are credited.
License URL: https://creativecommons.org/licenses/by/3.0/

Keywords: Oxygen, Hypoxia, Fluorescence, Photobiology, Coral-algae competition

Funding: United States National Science Foundation OCE-0927448 Air Force Office of Scientific Research Natural Materials and System and Extremophiles Program FA9550-10-1-0112 This research was supported by the United States National Science Foundation (NSF) award OCE-0927448 to Jennifer E. Smith, and by the Air Force Office of Scientific Research (AFOSR) Natural Materials and System and Extremophiles Program, under award FA9550-10-1-0112 to Dimitri Deheyn. The funders had no role in study design, data collection and analysis, decision to publish, or preparation of the manuscript.

==============================
While shifts from coral to seaweed dominance have become increasingly common on coral reefs and factors triggering these shifts successively identified, the primary mechanisms involved in coral-algae interactions remain unclear. Amongst various potential mechanisms, algal exudates can mediate increases in microbial activity, leading to localized hypoxic conditions which may cause coral mortality in the direct vicinity. Most of the processes likely causing such algal exudate induced coral mortality have been quantified (e.g., labile organic matter release, increased microbial metabolism, decreased dissolved oxygen availability), yet little is known about how reduced dissolved oxygen concentrations affect competitive dynamics between seaweeds and corals. The goals of this study were to investigate the effects of different levels of oxygen including hypoxic conditions on a common hermatypic coral Acropora yongei and the common green alga Bryopsis pennata. Specifically, we examined how photosynthetic oxygen production, dark and daylight adapted quantum yield, intensity and anatomical distribution of the coral innate fluorescence, and visual estimates of health varied with differing background oxygen conditions. Our results showed that the algae were significantly more tolerant to extremely low oxygen concentrations (2–4 mg L−1) than corals. Furthermore corals could tolerate reduced oxygen concentrations, but only until a given threshold determined by a combination of exposure time and concentration. Exceeding this threshold led to rapid loss of coral tissue and mortality. This study concludes that hypoxia may indeed play a significant role, or in some cases may even be the main cause, for coral tissue loss during coral-algae interaction processes.

Introduction

Over the last several decades coral cover has been declining around the globe (Gardner et al., 2003; Bruno & Selig, 2007). These losses in live coral cover are often associated with increases in the abundance of fleshy algae (Hughes, 1994; McCook, 1999). Multiple causes for these so called “phase shifts” have been identified including warming and subsequent coral bleaching and mortality or local anthropogenic influences such as pollution and overfishing (Bellwood et al., 2004; McManus & Polsenberg, 2004; Hughes et al., 2010). The underlying competitive mechanisms of coral-algae interactions still remain poorly understood (McCook, 2001; Jompa & McCook, 2003; Smith et al., 2006). However, the outcomes of these competitive interaction processes are critical for determining the resulting relative abundance of these organisms in coral reef habitats (McCook, 2001).

Both corals and algae use various physical mechanisms (e.g., sweeper tentacles, messentarial filaments, abrasion, shading) to compete with one another for the limited substratum available on the benthos (Nugues & Roberts, 2003; Van Veghel, Cleary & Bak, 1996; Coyer et al., 1993; Tanner, 1995). In addition, such competition may also involve direct chemical mechanisms, such as allelopathy (Bak & Borsboom, 1984), or the enhancement of disease transfer (Nugues et al., 2004) to other sessile organisms (McCook, Jompa & Diaz-Pulido, 2001). Furthermore, an indirect mechanism has also been suggested where organic carbon released by algae promotes microbial activity, which in turn can affect coral physiology and causes damage to the holobiont. The main driver of coral mortality in this proposed scenario is hypoxia resulting from intensified microbial activity and respiration at the coral/algal interface (Smith et al., 2006). There is increasing evidence that algal exudates can cause microbe-mediated coral mortality but more information is needed about the specific mechanisms involved (Barott et al., 2009).

It is known that oxygen concentrations well below saturation level can occur on coral reefs (Nilsson & Östlund-Nilsson, 2004; Wild et al., 2010). In such complex environments, in situ oxygen levels can fluctuate widely on a diurnal cycle and can be very low at night when organismal respiration dominates the landscape (Haas et al., 2010a; Wild et al., 2010). The extent of this diurnal fluctuation is thereby largely influenced by the biological nature of the local benthic community (Niggl, Haas & Wild, 2010). For example, dissolved oxygen (DO) concentrations in water surrounding algae-dominated areas can be lower than in adjacent coral-dominated reef areas, reaching DO concentrations as low as 4 mg L−1 (Haas et al., 2010a).

Benthic algae are known to release a significant portion of their daily fixed carbon as photosynthetates, primarily carbohydrates (Haas & Wild, 2010), thus enriching the immediate surrounding water with dissolved organic carbon (DOC) (Khailov & Burlakova, 1969; Brylinsky, 1977; Haas et al., 2010a; Haas et al., 2010b). It is also known that water column microbial communities consume some of this DOC. As a result of this carbon uptake (Nelson et al., 2011; Haas et al., 2011) increased microbial metabolism and concomitant consumption of oxygen via respiration (Wild et al., 2010; Haas et al., 2010a), can result in locally confined (from the mm to cm spatial scale) but sometimes severe hypoxic conditions (Barott et al., 2009). These hypoxic conditions are likely most severe at night when all taxa are consuming oxygen via respiration but recent studies have shown that increased biological oxygen demand may even outweigh photosynthetic oxygen release during daylight (Haas et al., 2013). Clearly the characteristics of coral algal interaction zones will depend upon the taxa present (their species and relative abundance) and their respective metabolic rates. Despite this variability, controlled laboratory experiments have shown that these hypoxic conditions can be reversed with the addition of antibiotics suggesting that bacteria play a key role in these interactions (Smith et al., 2006).

There is thus strong evidence that hypoxia plays an important role in coral-algae competition and therefore on processes structuring coral reef communities (McCook, 2001). However, much less is known about how corals and algae individually respond to reduced oxygen conditions. Assuming that algal exudates can fuel microbial community metabolism, resulting in hypoxic conditions at the coral — algal interface, algae need to be more tolerant to these hypoxic conditions than corals to be the competitive superior. The goals of the present study were to investigate the differential tolerance and responses of a common Indo-Pacific coral and a green macroalga to reduced DO conditions in independent incubation experiments.

Material and Methods

This study was conducted in controlled laboratory conditions at the Scripps Institution of Oceanography (SIO) in San Diego, California from July 17 to 27, 2011. The hermatypic coral Acropora yongei initially provided by the Birch Aquarium at Scripps in 2010 has been maintained in culture under optimal growing conditions at Marine Biology Research Division Experimental Aquarium Facility at SIO (Roth et al., 2010). The Birch Aquarium at Scripps also supplied specimens of the green alga Bryopsis pennata specifically for this study. All samples were grown in controlled environments, allowing for better comparison amongst treatments, which is somewhat simplified and possibly not thoroughly reflecting variation and adaptation likely found in-situ. Organisms were kept in identical flow-through tanks with filtered (nominal pore size 50 µm) and temperature controlled seawater at least 10 days prior to the experiment (from now on referred to as “seawater”). Seawater temperature was monitored at 5 min intervals with data loggers (Onset HOBO® Pendant UA-002-64) and was 26.3 ± 0.4°C for the duration of the experiment.

Adequate irradiance was provided to the corals and algae by artificial illumination (2 × 54 W 6000 K Aquablue+, 1 × 54 W 6000 K Midday, and 1 × 54 W Actinic+, Geismann, Germany) placed 80 cm above the surface water level with a daylight cycle of 12–12 h. The resulting photosynthetically active radiation (PAR) was 120 µmol quanta m−2 s−1, as measured with a LICOR LI-193 Spherical Quantum Sensor.

Sample preparation

Corals were carefully fragmented into individual branches (∼5 cm in length) with bone cutters. The coral fragments were then attached to ceramic tiles (3.6 × 3.6 × 1.4 cm) with coral cement (HoldFast®, Instant Ocean). A polyethylene holding stick was also attached to the ceramic tile to reduce direct mechanical stress on the organisms during experimental handling, all following a process routinely done in our laboratory (Roth et al., 2010).

Algal specimens were attached to identical tiles using small plastic zip ties. Sizes of all specimens were chosen in a way that both organism groups would have comparable oxygen consumption rates of ∼2 mg L−1 during the 12 h dark incubation period (as determined in preliminary analysis; data not shown). All specimens were allowed to recover from fragmentation for 4 or 2 weeks for corals and algae, respectively.

Experimental set up

During each 12 h daytime phase all specimens (n = 18 corals, n = 18 algae) were kept in their tanks under identical conditions. At the beginning of nighttime, specimens were each incubated in separate airtight containers (1 L mason jars) and subjected to 3 different (n = 6 for each organism group) dark oxygen concentration treatments: 6–8 mg L−1 (ambient oxygen concentrations), 4–6 mg L−1 (decreased oxygen concentration), and 2–4 mg L−1 (low oxygen concentration). Oxygen concentrations were chosen to reflect oxygen concentrations in well mixed, more exposed reef environments (our “ambient” equivalent), more sheltered backreef or low water flow reef environments (our “decreased” equivalent) and oxygen concentrations found at coral-algae interfaces or coral interstices (our “low” equivalent). Reduced dissolved oxygen concentrations were generated by sparging seawater with nitrogen to varying levels prior to the start of the nighttime experimental treatment, while simultaneously measuring dissolved oxygen concentrations with optical oxygen probes (HACH LANGE HQ40; precision 0.01 mg L−1, accuracy ± 0.05%) to reach the desired oxygen concentration.

The airtight experimental containers were equipped with stir-bars to ensure constant water circulation and were kept in the dark in a temperature-controlled environment (25.9 ± 0.7°C) for 12 h. At the end of the nocturnal oxygen treatments, specimens were removed from their individual containers and again placed in the daytime maintenance tanks with ambient oxygen concentrations. Seawater from experimental nighttime containers was then rapidly analyzed for dissolved oxygen concentrations to ensure comparable oxygen consumption amongst treatments. This procedure was repeated over 10 consecutive diurnal cycles for all 36 specimens, even when showing clear signs of stress (i.e., bleaching and/or tissue loss).

Under such an experimental set up, oxygen consumption during the dark incubations was similar between algae and coral specimens and on average 0.141 ± 0.006 mg L−1 h−1 and 0.144 ± 0.006 mg L−1 h−1 (mean ± SE) in the respective incubation setups. Both coral and algae exposed to low oxygen treatments (2–4 mg L−1) showed significantly higher oxygen consumption rates (F = 5.21, p < 0.001) compared to the algae in decreased oxygen treatments (4–6 mg L−1) or ambient oxygen concentration treatments (6–8 mg L−1) (Table 1).

Table 1 Nighttime oxygen variations.

Dissolved oxygen (DO) concentrations (mean ± standard error) measured at start and end of nighttime oxygen treatments along with the corresponding nighttime oxygen consumption rates. Rate values with different letters are significantly different (one-way ANOVA followed by Tukey post hoc tests; α = 0.05).

Treatment	DO start (mg L−1)	DO end (mg L−1)	O2 draw down (mg L−1 h−1)	
Algae 6–8 mg L−1	7.99 ± 0.01	6.45 ± 0.08	0.13 ± 0.007 (A)	
Algae 4–6 mg L−1	5.99 ± 0.00	4.51 ± 0.14	0.12 ± 0.011 (A)	
Algae 2–4 mg L−1	4.01 ± 0.01	1.93 ± 0.14	0.17 ± 0.011 (B)	
Corals 6–8 mg L−1	7.99 ± 0.01	6.45 ± 0.06	0.13 ± 0.005 (A)	
Corals 4–6 mg L−1	5.99 ± 0.00	4.34 ± 0.16	0.14 ± 0.013 (A, B)	
Corals 2–4 mg L−1	4.01 ± 0.01	1.98 ± 0.08	0.17 ± 0.007 (B)	

Biological parameters measured

Overall health of corals and algae was assessed separately using a combination of measurements on each sample. Photosynthetic oxygen production assessments and PAM measurements were used to address changes in the photosynthetic performance of the specimens and photographic analysis was used to assess the general health of coral and the growth rates of the algae (detailed description below). All described measurements were conducted on day −3, 0, 1, 3, 7, and 10 of the experiment.

Changes in oxygen production during photosynthesis were assessed by placing each specimen in an individual, temperature-controlled and gently stirred (∼90 rpm) beaker during daylight hours (between 1000 and 1230 PST). Initial DO readings were obtained from each beaker using the above mentioned oxygen optodes. Beakers were then sealed airtight and placed back under artificial daylight conditions; a final DO reading was taken after 150 min. The difference between the initial and final DO readings was used as the net photosynthetic oxygen production rate of each specimen. After these measurements the organisms were then placed back in their respective maintenance tanks.

Pulsed Amplitude Modulation (PAM) fluorescence was used to assess the photosynthetic efficiency of the organisms (Ralph, Gademen & Dennison, 1998). The photochemical efficiency of PSII was evaluated by measuring the quantum yield (QY) derived from the Fv/Fm values obtained from the PAM, where Fv is variable fluorescence and Fm is maximum fluorescence of the measurements. PSII fluorescence measurements were conducted following saturating light pulses (800 ms flashes of 8,000 µmol quanta m- 2 s- 1 PAR). In this study we assessed dark-adapted (or maximum) QY during early pre-dawn hours (analyses from 07:00 to 08:00 PST) and daylight adapted (or effective) QY during mid-day (analyses from 14:00 to 15:00 PST), after 6 h of full light exposure (Genty, Briantais & Baker, 1989).

Digital image analysis was systematically performed under identical conditions for all samples throughout the experimental period. Green fluorescence was hereby used as a general indicator for coral health, while red fluorescence was used as estimation for functional chlorophyll abundance. Images from corals and algae submerged in seawater were collected using a Nikon SMZ 1500 microscope equipped with an epi-fluorescence setup and X-cite 120 (EXFO, Lumen Dynamic) mercury lamp for excitation light source (see Roth et al., 2010 for details). A RETIGA 2000R QImaging (photometrics) digital camera was used to take the pictures through computer controlled Q-Capture Pro software. For each day of analysis, calibration images of a white ruler with fluorescence in the green were taken with identical settings in order to allow for detection of potential drift from the instrumentation. All images (tiff) of organisms were then analyzed in Matlab 7.5 (Mathworks Inc., Natick, MA, USA) as described in Roth et al. (2010).

Specimens were photographed under full light spectrum of a Dolan-Jenner 180 Fiber-Light High Intensity Illuminator and then directly under fluorescence (exc. 450–490 nm); note that data for the fluorescence at day 0 for one coral was lost, resulting in lower replication for some of the analyses based on changes from day 0 to 10. For the coral, white light images were used to calculate the ratio of pixels showing white color (90% saturation) versus the entire number of pixels. White light images were used to calculate surface area of the specimens. The surface area was then used to calculate the specimen’s average green fluorescence intensity by subtracting the average background pixel intensity from the green pixel intensity, which was then summed and divided by the surface area. White light images were also used for a qualitative visual evaluation of coral and algae health, with special attention on changes in size, color, lesions, and/or loss of tissue. The flexible and overlapping nature of the algal specimens did not allow for precise quantification; yet images were used to estimate relative algal growth by quantifying the red fraction of pixels (in fluorescence, thus representing chlorophyll) from each of the respective images over the duration of the experiment.

Statistical analysis

All data (oxygen production, Maximum QY, Effective QY, Red fluorescence fraction, Green fluorescence intensity, Green fluorescence fraction) were log-transformed data to meet assumptions of normality and homocedasticity. Effects of different oxygen treatments on the coral and algal samples were tested for significance by comparing changes in a variety of parameters from experimental day 0 to 10 only. The comparison tested the difference in the magnitude of change from start to end of the experimental period for a given treatment using one-way analysis of variance (ANOVA), and differences among oxygen treatments were assessed with Tukey post hoc tests (α = 0.05). For all measurements conducted on both coral and algae a two-way ANOVA with oxygen treatment (3-levels) and species (2-levels) was used (which thus excludes the fluorescence analyses that were different for each species). Comparisons between light and dark-adapted QY measurements conducted for each specimen were performed using a paired sample T-test. Results showing the main effects and the interactions are given in Table S1. These statistical evaluations were performed using SAS within the software package JMP (v9; SAS institute 1989–2011).

Additionally a Repeated-Measures-ANOVA was used to incorporate in the analysis the percent of change found in the data that relates to the evolution over time of individual coral and algal samples repeatedly measured during their exposure to different treatment conditions. The factors in this analysis include the experimental oxygen treatment, the time point, the repetition of measurements from the same samples (Individual factor) and its variation over time (Repetition × Time factor). The percent of variance explained by each factor (or combination of factors) on the measurements made from the corals and alga was based on the relative expression of the Sum of Squares of each factor to the total of the Sum of Squares. The percent of variance remaining unexplained was listed as a residual factor and analyses were conducted with the statistical software Statview® 5.0 (SAS Institute, Inc.).

Results

Overall, the different oxygen treatments significantly affected various metrics of coral health while no significant impacts were found on algal health and/or physiology. This was clear in particular for the fluorescence analyses that were, however, specific to each organism (see specific sections below). When considering the analysis common to both the coral and the algae, the effects of oxygen depletion treatment were greater for the maximum quantum yield, while the effective quantum yield and the photosynthetic oxygen production were less affected (Table S1). Maximum quantum yield appeared affected significantly by the oxygen treatment and species, as well as their interaction; thus the oxygen depletion had effects that were different between coral and algae with regards to the maximum quantum yield (species*oxygen treatment of Max. QY; Table S1). As for the effective quantum yield, it showed significant difference between organisms, but in both species this parameter showed no significant variation with the oxygen treatment (whether oxygen treatment or species*oxygen treatment; Table S1). As for the photosynthetic oxygen production, it was not significantly different between organisms, yet was a parameter sensitive to oxygen treatment especially when considering it within species (species*oxygen treatment; Table S1).

Photosynthetic oxygen production measurements were not different over the course of the experiment for all algae treatments and for corals in the ambient (6–8 mg L−1) and decreased (4–6 mg L−1) oxygen treatment. Corals in the low oxygen treatment (2–4 mg L−1) showed a significant decline in oxygen production rates (52.7 ± 12.7% of the initial values) over the experimental period (Fig. 1.1, Table 2). All algae displayed an initial tendency to decrease on the first experimental day but recovered from day one onward, and reached their initial levels again by the end of the experimental period (Fig. 1.1A).

Figure 1 Summary of responses from the coral A. yongei and alga B. pennata to the oxygen treatments.

(1.1) Photosynthetic oxygen production, (1.2) pulse amplitude modulation fluorescence measurements of maximum (dark-adapted) quantum yield, (1.3) fraction of red pixels in images of algal fluorescence, (1.4) green intensity, (1.5) fraction of green pixels, and (1.6) red intensity of images taken from coral fluorescence. Panel A shows mean values (± standard error) and a corresponding derived simple spline curve for each treatment over the 10 d experimental period. Panel B–E show box plots (data range and mean value indicated by black line) of the deviation to initial values for each measurement day (B = day 0–1, C = day 0–3, D = day 0–7, E = day 0–10).

Table 2 One-way ANOVA of the effect of oxygen treatments on the biological parameters.

Statistical analysis of the effect of each oxygen treatment on the biological parameters measured from algae and corals (one-way ANOVA). Significance of treatment on the effect was tested on the difference in values of each parameter between experimental day 0 and 10 (Tukey post hoc tests). Treatments with different letters indicate significant differences (α = 0.05) in the changes of the respective parameter.

Group	O2
treatment	O2 production	Maximum QY	Effective QY	Green intensity
× fraction	Red intensity	
		ANOVA	Tukey	ANOVA	Tukey	ANOVA	Tukey	ANOVA	Tukey	ANOVA	Tukey	
Algae	6–8 mg L	F(2, 15) = 0.09,
p = 0.918	A	F(2, 15) = 0.94,
p = 0.411	A	F(2, 15) = 0.66,
p = 0.530	A	*	*	*	*	
4–6 mg L	A	A	A	
2–4 mg L	A	A	A	
Coral	6–8 mg L	F(2, 15) = 3.49,
p = 0.032	A	F(2, 15) = 12.63,
p < 0.001	A	F(2, 15) = 4.29,
p = 0.037	A, B	F(2, 14) = 39.73,
p < 0.001	A	F(2, 15) = 5.76,
p = 0.014	A	
4–6 mg L	A	A	A	B	A, B	
2–4 mg L	B	B	B	C	B	
Notes.

* Green fluorescence was not measured for algae and red fluorescence was only considered as the fraction of red pixels used as a proxy for size.

PAM measurements. Over the course of the 10-day experiment dark-adapted QY values recovered for all specimens to pre-experimental values, except for corals subjected to the low oxygen treatment. In this treatment the dark-adapted QY values gradually declined until they reached ∼70% of their initial values on day three and stayed at this level thereafter (Figs. 1.2B–1.2E). Decreases in dark-adapted QY values of corals subjected to low oxygen treatment were significantly greater at the end of the experiment than of any other coral treatment (Tukey p < 0.05) (Fig. 1.2E, Table 2).

Daylight adapted QY measurements followed the same pattern as the dark-adapted QY (data not shown). After a slight initial decrease, daylight adapted QY recovered in all treatments to pre-experimental conditions, except in the low oxygen coral treatment, which stayed at ∼80% of the initial value. This treatment showed significantly greater decreases than all other treatments.

Daylight and dark-adapted QY values were always in the same range and, with one exception, dark-adapted QY values were always higher for the same specimen on the same day as daylight adapted QY. Only on day 10 in the 2–4 mg L−1 oxygen treatment all coral specimens showed significantly higher daylight adapted QY than the dark-adapted QY values (paired sample T-test; p = 0.038; Fig. S1).

Fluorescence analysis of algae showed significantly higher fraction of red pixels at the end of the experiment for all treatments (F = 27.46, p < 0.001), yet with no detectable difference in the rate of increase between the respective oxygen treatments (Fig. 1.3). Although there was an initial drop in the fraction of red pixels, which was similar with all parameters measured for algae (Figs. 1.3A–1.3C), a strong increase in the fraction of red pixels in algae samples from day 3 onward suggests a constant growth or increase in chlorophyll concentration for these specimens (Figs. 1.3A, 1.3D and 1.3E)

Coral fluorescence analysis revealed significant decreases in green intensity between the 2–4 and 4–6 mg L−1 treatments compared to the 6–8 mg L−1 control treatment at the end of the experiment (F(2,14) = 4.60, p = 0.029) (Fig. 1.4E, Table 2). Green fluorescence intensity values for the 6–8 mg L−1 control treatment stayed in the same range throughout the experiment (Fig. 1.4A). The fluorescence intensities in the 4–6 mg L−1 treatment gradually declined over the course of the experiment, while in the 2–4 mg L−1 treatment they decreased rapidly to the point of not being detectable after day 3 (Figs. 1.4A–1.4E). Because intensity measurements were calculated for each individual pixel on the green channel, we also determined whether the number of pixels in the green channel changed, representing the area of fluorescence in the samples. While the area of green fluorescence remained similar for corals in the 6–8 mg L−1 control treatment over the duration of the experiment, there was a gradual decrease in the 4–6 mg L−1 treatment; the area of green fluorescence was about 3× smaller than the control by the end of the experiment (Figs. 1.5A, 1.5C–1.5E). The fraction of green pixels in corals showed significantly greater decreases in the 2–4 mg L−1 oxygen treatment than in the 4–6 and 6–8 mg L−1 treatments by the end of the experiment (F(2,15) = 63.21, p < 0.001), being about 10× smaller than in the controls (Figs. 1.5A–1.5E). The fraction of green fluorescent pixels rapidly declined within the first three experimental days (Fig. 1.5A) and was barely detectable by the end of the experiment.

The intensity of fluorescence and the amount of pixels producing green fluorescence are inherently correlated and they were thus multiplied in order to obtain one single combined “green fluorescence parameter” (Fig. S2). This parameter was significantly different amongst corals from all three oxygen treatments at the end of the experiment (Table 2).

In contrast, the increase in the signal of coral red fluorescence in the 2–4 mg L−1 (Figs. 1.6A–1.6C) was significantly higher than in the 6–8 mg L−1 control treatment at the end of the experiment (Table 2). The 4–6 mg L−1 treatment showed slight increases in the intensity of red pixels over the experimental period, but values were not significantly different from those of either of the other treatments at the end of the experiment (Figs. 1.6D and 1.6E; Table 2).

Visual census indicated that all corals subjected to low (2–4 mg L−1) oxygen treatments had lesions and partial tissue loss within the first 3 days and all specimens were dead or dying, with no coral tissue covering the calcareous coral skeleton, after day 3 (Fig. 2). These specimens rapidly became discolored (in bright field) instead of staying white, likely the result of a microbial film and/or endolithic or micro-algae growing in/on the skeleton. No other experimental specimens, under any of the other treatments showed visually detectable signs of stress or tissue loss during the course of the experiment.

Figure 2 Fluorescence timeline.

Representative images of the coral A. yongei and the alga B. pennata in bright field (upper panels) and fluorescence (lower panels) for two different oxygen treatments over the time of the experiment. (A) coral 6–8 mg L−1, (B) coral 2–4 mg L−1, (C) alga 6–8 mg L−1, (D) alga 2–4 mg L−1.

Repeated-measures-ANOVA showed that algae and corals had distinct responses to the various oxygen treatments, but also that these responses were primarily driven by different factors (Table S2). The algae showed little response to the oxygen treatments (responsible for 0.6% of changes in red fluorescence, and up to 12.5% for O2 production). The majority of effects observed in algae were driven by the Time factor (ranging from 14.4% for the Effective QY to 47.5% for the Maximum QY) and the Residual factor (ranging from 31.1% for the maximum QY to 50.5% for the Effective QY) (Table S2).

The influence of these factors was differently distributed for corals. The Treatment factor contributed from 14.1% (for daylight oxygen production rates) to 41.0% (for green fluorescence fraction) of the observed variability and was always (except for O2 production) a statistically significant factor (Maximum QY: F = 9.998, p = 0.0017; Effective QY: F = 8.386, p = 0.0036; Green fluorescence intensity: F = 8.699, p = 0.0035; Green fluorescence fraction: F = 33.453, p < 0.0001; Green fluorescence intensity * fraction: F = 21.752, p < 0.0001; Red fluorescence intensity: F = 17.223, p = 0.0001). The Time factor contributed less to the variability but was still important, with contribution ranging from 9.2% (for green fluorescence fraction) to 37.6% (for daylight oxygen production rates). These results indicate that for the corals, daytime oxygen production rates were the least affected parameter after repeated exposure to nighttime hypoxia, while the fraction of green fluorescence was the most affected. Repetition factor effects ranged from 1.5% for O2 production to 15.6% for the Maximum QY, and from 15.6% for the red fluorescence fraction to 39.3% for the Green fluorescence fraction. Such a large effect was also observed for the red fluorescence of algae (37.7%) thus suggesting that measuring fluorescence from complex-shaped samples (such as organisms) as a strong inherent variability due to repetition that needs to be taken into account. This variability also evolved with time since the Repetition * Time factor was often significant, ranging from 10.7% for the Green Fluorescence intensity to 22.4% for the Green Fluorescence fraction, while ranging from 9.3% for oxygen production to 23.3% for Effective QY. As for the Residual factor, it was usually lower in corals than for algae, ranging from 4.6% for the Green Fluorescence fraction to 37.5% for the O2 production (Table S2).

Discussion

Here we examined the response of a common species of coral and reef algae to experimentally manipulated oxygen conditions reported to occur at night in coral reef environments, where DO can decrease dramatically in certain areas, due to organismal respiration and microbial metabolism. Hypoxia, defined as dissolved oxygen concentrations below 2 mg L−1 (Stevenson & Wyman, 1991) has been repeatedly described in coral reef ecosystems, especially within interaction zones or reef interstices (Shashar, Cohen & Loya, 1993; Barott et al., 2009). Hypoxia has also been hypothesized to play a role in coral mortality at the location of coral algal interaction zones (Smith et al., 2006; Barott et al., 2009). This study demonstrates for the first time that low oxygen concentrations can have deleterious effects on the health and physiology of a hermatypic coral, Acropora yongei, while causing little to no effect on the common green alga Bryopsis pennata.

Physiological effects of decreased oxygen concentrations

While the alga showed no measurable response to exposure to the different oxygen, the hermatypic coral A. yongei displayed significant alterations in all parameters measured, when subjected to the low oxygen (2–4 mg L−1) treatment. The samples in these treatments, without exception, appeared bleached, lost major portions of their tissue and most likely were deceased within 3 days of the experiment. The visible decline in coral health was accompanied by a significant decrease in photosynthetic performance, which was assessed via oxygen production rates, dark-adapted maximum quantum yield, and daylight adapted effective quantum yield measurements. Surprisingly, photosynthetic activity of these low oxygen treatment coral specimens was still measurable on experimental day 3 and remained at a constant level of 50–80% of the initial measurements thereafter. This could either have been related to remaining zooxanthellae (though unlikely), endolithic algae, and/or colonization of the bare coral skeleton by cyanobacteria or microalgae. The latter is supported by the observation of tissue loss and, subsequently, a greenish coloration that the bleached coral skeleton rapidly acquired. The photosynthetic organisms associated with the low oxygen coral samples at the end of the experiment showed higher values of effective than maximum QY. This exception might stem from a fast regeneration of their photosystems after the cessation of the nocturnal hypoxic stress conditions, which potentially compromised the photosynthetic performance of these organisms.

The visual census showed that the coral tissue in low oxygen treatments dissolved within the initial 3 days. This clearly attributes all photosynthetic performance measured after day 3 to endolithic algae and/or microalgae rapidly colonizing the coral skeleton. It further indicates that the bare coral skeletons were immediately colonized by microalgae, capitalizing on the new hard substratum, even though parts of the colony were still covered with residual coral tissue on experimental day 1–3.

Green fluorescence

Physiological measurements were supported by results obtained from analysis of fluorescence images. The scleractinian coral A. yongei used in this experiment produces proteins that fluoresce exclusively in green (GFPs) (Roth et al., 2010; Roth, Goericke & Deheyn, 2012). GFPs are ubiquitous in scleractinian corals (Alieva et al., 2008; Gruber et al., 2008; Salih et al., 2000) and can constitute a significant portion of their total protein content (Leutenegger et al., 2007). Although there is currently no consensus on the physiological function/s of these proteins in corals, previous studies have demonstrated the potential to use GFPs as an indicator of health of the organism (Roth & Deheyn, 2013). Here the green fluorescence was significantly affected in the corals subjected to the low oxygen treatments. There were also significant responses detectable by fluorescence analysis for coral specimens subjected to 4–6 mg L−1 nighttime oxygen concentrations. Noticeable decreases in the intensity of green pixels were accompanied by a slight decrease in the extent of the fluorescent area (thus decrease in number of green pixels). We combined the GFP responses to the different treatments by multiplying the green fluorescence intensity (intensity of pixels with fluorescence) by the fraction of the green signal (number of pixels with fluorescence) in each picture (i.e., intensity * abundance). The combination of both parameters incorporates changes of the fluorescence intensity from the proteins themselves as well as the difference in spatial distribution of the fluorescence (coenosarc and polyps versus mainly polyps). This spatial difference of the green fluorescence that becomes mainly visible around the polyps can be interpreted as the result of (1) the green fluorescence is increasingly shaded by the increasing amount of zooxanthellae, especially in the coenosarc (see following section), and/or (2) the GFP could be denatured following some biochemical reaction such as free-radical chelation, (Bou-Abdallah, Chasteen & Lesser, 2006; Palmer, Modi & Mydlarz, 2009) which could be more pronounced in the coenosarcs (Fig. S1). The clear detection of subtle changes in corals subjected to the decreased oxygen treatment suggests that coral fluorescence may be an extremely sensitive indicator to assess overall coral health (Roth et al., 2010; Roth, Goericke & Deheyn, 2012; Roth & Deheyn, 2013) and further emphasizes the value and sensibility of fluorescence analytical methods in determining early stage changes in coral health conditions (see also Table S1).

Red fluorescence

Fluorescence signals can also be used to distinguish between pigmented and bleached corals and between coral and algae (Myers et al., 1999). Emission in red, at wavelengths longer than 630 nm, is not generated by the coral host (Mazel, 1995; Salih et al., 2000; Neori et al., 1988), but from photosynthetic pigments found in their algal symbionts (Jeffrey & Haxo, 1968), which show a distinct chlorophyll peak at 680 nm (red light) when excited with blue light (480 nm) (Gurskaya et al., 2001). The detected rise in red fluorescence for corals in low oxygen treatments may therefore either be attributed to increases in zooxanthellae (yet unlikely since the corals were bleached and dying), or to the growth of endolithic algae (already present in the coral skeleton), cyanobacteria and/or filamentous algae starting to colonize the bare coral skeleton (Shashar et al., 1997; Zawada & Jaffe, 2003).

While green fluorescence was more visible in the polyp tissue, red fluorescence, initially generated by the endosymbiotic zooxanthellae, was mainly visible in the coenosarcs of the coral fragments. This particular compartmentalization is not a general feature for the species A. yongei (Roth et al., 2010) as the difference in anatomic distribution of coral pigments and associated zooxanthellae can be dynamic and vary within and between coral species (Gruber et al., 2008; Oswald et al., 2007). However, the strong decrease and eventual extinction of green fluorescence (when compared to background values), which was paralleled by a uniform increase in red fluorescence of corals subjected to low oxygen concentrations, suggests the growth of endolithic and/or opportunistic filamentous algae in/on the coral skeleton rather than an increase in symbiontic zooxanthellae.

In contrast to the corals, the area of red fluorescence generated by the photosynthetic pigments of the alga Bryopsis p. increased gradually over time in all treatments, indicating similar growth of all algae samples regardless of the nighttime oxygen concentration treatments.

Conclusion

Although extremely low oxygen concentrations (2–4 mg L−1) had severe impacts on A. yongei specimens over a short period of time (Fig. 1; Table S1), we also show that corals were able to tolerate reduced oxygen concentrations reasonably well (within 4–6 mg L−1). Nighttime oxygen concentrations between 4 and 6 mg L−1, which are commonly found during the early morning hours in various reef locations (Kraines et al., 1996; Niggl, Haas & Wild, 2010; Wild et al., 2010; Haas et al., 2010a), showed some effects on fluorescent proteins, but not on the physiological performance of the respective corals over the experimental period of 10 days. A surplus of oxygen consumption to naturally occurring low nighttime DO concentrations, as may be facilitated by proximate algae in combination with algal exudate induced increases in microbial oxygen consumption, may however exceed the tolerance (< 4 mg L−1) and lead to rapid tissue loss and colonization of the calcareous coral structure by algae (Fig. 2). Additional evidence on the importance of this interaction mechanism is given by studies conducted by Barott et al. (2009), who showed that oxygen concentrations on the interfaces of coral algae interaction zones were on average 3.2 ± 0.5 and 2.9 ± 0.4 mg L−1 when algae were the superior competitors (Pocillopora verrucosa vs. mixed red turf algae and Montipora sp. vs. mixed red turf algae, respectively); an oxygen concentration which is just below the coral tolerance threshold suggested by the present study, but not harmful to the algae we studied. The study further revealed that for coral algae interaction zones, where both organisms were in a stable state or corals were the superior competitor (Favia sp., Montipora sp. and Pocillopora sp. vs. crustose coralline algae), oxygen concentrations on the interfaces were on average 7.9 ± 0.7 mg L−1, i.e., well above the suggested threshold of 4 mg L−1.

Finally, our study presents two main findings: (1) the alga Bryopsis pennata was significantly more tolerant than the hermatypic coral Acropora yongei to extremely low oxygen concentrations (<4 mg L−1), and (2) the coral could tolerate decreased oxygen concentrations up to a given point. The threshold (∼4 mg L- 1) is below reported diurnal oscillations facilitated by the coral reef community metabolism, but lies within oxygen concentrations reported from the interface along coral algae interaction zones. Beyond this threshold the corals used in this study experienced rapid loss of tissue and death of the whole organism. This study therefore suggests that hypoxia may be a factor influencing competitive interactions between the reef-building corals such as A. yongei and common benthic algae such as B. pennata. Further research will investigate whether such mechanisms could be generalized to other species of corals and algae.

Supplemental Information

Table S1 Two-way ANOVA of oxygen treatment and species effects

Results from two-way ANOVA analysis with the main effects of oxygen treatment (3-levels) and species (2-levels) and their interaction on data measured using both coral and algae (fluorescence analyses were excluded of this analysis because not shared by both species). Statistically significant effects are shown in bold.

Click here for additional data file.

Table S2 Contribution of each factor on changes in organism health

Contribution (%) that each of the studied factors (oxygen treatment, experimental time, individual repeat, and individual repeat variation over time) has on the metrics of organism health measured from the coral and algae. The residual indicates fraction of the contribution that remains unexplained by the present factors. Each factor is associated with a different degree of freedom (DF); statistically significant effects are shown in bold (p < 0.05) and marked with an asterisk when p < 0.01 (from Repeated Measures ANOVA).

Click here for additional data file.

Figure S1 Quantum yield values

Maximum and effective quantum yield values (QY) for coral specimen subjected to the low oxygen treatment over the course of the experimental period. Individual coral specimens are marked by color while the bold black lines indicate the average effective (dotted lines) and maximum (solid lines) QY values. Note that only on experimental day 10 effective QY values are always higher than maximum QY values. Lines represent a locally weighted scatterplot smoother, i.e. Kernel Smoother.

Click here for additional data file.

Figure S2 Coral low oxygen treatment fluorescence

Representative images of the coral A. yongei subjected to (A) 6–8 mg L−1, (B) 4–6 mg L−1, and (C) 2–4 mg L−1 in fluorescence. The pictures visualize the changes in brightness (i.e. amount of GFPs) and distribution (i.e. coenosarc and polyps versus polyps mainly) of green fluorescence.

Click here for additional data file.

We thank the Hughes Scholar Summer Internship Program for giving Melissa Thompson the opportunity to conduct this work as a research fellow in the Deheyn Laboratories.

Additional Information and Declarations

Competing Interests

Author Contributions

The authors declare that there are no competing interests.

Andreas F. Haas conceived and designed the experiments, performed the experiments, analyzed the data, wrote the paper.

Jennifer E. Smith and Dimitri D. Deheyn conceived and designed the experiments, analyzed the data, contributed reagents/materials/analysis tools, wrote the paper.

Melissa Thompson performed the experiments, analyzed the data, wrote the paper.

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
