# Peer review of "Effects of reduced dissolved oxygen concentrations on physiology and fluorescence of hermatypic corals and benthic algae"

_PeerJ, doi:10.7717/peerj.235_

## Round 0.1 · original submission · Minor Revisions

Dear Authors,

First of all, our sincere apology for the unusually slow review process, which was due to the circumstances outside of our control.

We have now two reviews in hands, which are very positive but also raise some issues that require your attention and revision. Please go through the comments and submit your revised ms for further consideration.

Reviewer 1 ·

Basic reporting

No comments.

Experimental design

No comments.

Validity of the findings

No comments.

Additional comments

This study compared the effects of reduced oxygen concentrations on the physiology and fluorescence of a coral Acropora yongei and a green alga Bryopsis pennata. The rationale of the study is clearly spelled out. The experimental design is well thought out with the experiments being executed cautiously. Data are analyzed by appropriate statistical tests and results have provided convincing evidence on the differential responses of the coral and alga to reduced oxygen concentrations. The breadth and depth of the discussion is appropriate. The paper can be accepted for publication subject to minor revision. Below are comments and suggestions for improving the manuscript:

1) Line 168, replace “at” by “on”.
2) Line 280, Fig. S1 is not included in the submitted manuscript.
3) Line 307, Fig. S2 is not included in the submitted manuscript.
4) Lines 391-392. It should be moved to the Materials and Methods as nothing is mentioned in the Materials and Methods about why green fluorescence was measured.
5) Line 419-420. It should be moved to the Introduction.
6) Line 461. Replace “studies” by “studied”.
7) Fig. 1. Provide more information of the box plots. I guess the size of the box is the data range and the line in the box is the mean value.
8) Fig. 1. For results on corals under different oxygen concentrations, the colour contrast of the bars should be improved.
9) Table. 2. The data arrangement is wrong. The statistical results do not align with the column titles correctly.

Reviewer 2 ·

Basic reporting

No comments

Experimental design

Well done.

Validity of the findings

No flaw

Additional comments

Review of Peerj-868

This is interesting coral/algal interactions study where authors have successfully tested the effect low dissolved oxygen on their physiological health. This manuscript is timely contribution in the field of coral biology and marine ecology. The experimental design and data analysis used in the study are appropriate to address their objectives and there was no flaw. The results and the subject discussion are appropriate to publish in PeerJ. Although the manuscript is well written, need a revision before consideration for publication. I have the following suggestions for authors, which I hope would help them to revise the manuscript.

Abstract: It read well. “Amongst various potential mechanisms, algal exudates can mediate increases in microbial activity, leading to localized hypoxic conditions which could cause coral mortality in the direct vicinity” – is this is authors hypothesis? If so, it should be explicitly stated in the end whether is accepted or rejected. “extremely low oxygen concentrations (< 4 mg L-1)” – not clear, <4 is too general and authors should be specific – what happens at hypoxia level (2.8 mg?).


Introduction:
Again reads very well. Line 101: Authors need to explain why oxygen released during algal photosynthesis does not raise DO level in algal/coral interface?
Line 102-109: Need a solid hypothesis – authors have it (as shown in the abstract) and it should spell out clearly here.

Materials and methods

114 -115: The key point is that: animals were acclimatized in the laboratory for year or so – how could then results obtained from this study applicable to field condition – what are the limitations of this approach – this should be briefly stated here.

119: “prier” – how long?
145-150: Why these DO level?
166: Somewhere authors should say how these measured biological parameters will help them to address their question or hypothesis – I am not sure yet.

Results

Table 1: Looks too difficult to read – it can be easily converted into chart showing variation of DO during the experiment with SD values

Table 2: Looks an unusual ANOVA table – has to be modified using the standard Stat result tables.

Discussion
Well written

---

## Round 0.2 · accepted · Accept

I am happy with your revision and happy to accept your ms for publication as it is.